# The Effect of Travel-Chain Complexity on Public Transport Travel Intention: A Mixed-Selection Model

**DOI:** 10.3390/ijerph20054547

**Published:** 2023-03-03

**Authors:** Yuan Yuan, Chunfu Shao, Zhichao Cao, Chaoying Yin

**Affiliations:** 1Key Laboratory of Integrated Transportation Big Data Application Technology in Transportation Industry, Beijing Jiaotong University, Beijing 100044, China; 2School of Transportation and Civil Engineering, Nantong University, Nantong 226000, China; 3College of Automobile and Traffic Engineering, Nanjing Forestry University, Nanjing 210037, China

**Keywords:** public transport environment, trip-chain complexity, PLS-SEM, mixed-selection model

## Abstract

With urban expansion and traffic environment improvement, travel chains continue to grow, and the combination of travel purposes and modes becomes more complex. The promotion of mobility as a service (MaaS) has positive effects on facilitating the public transport traffic environment. However, public transport service optimization requires an accurate understanding of the travel environment, selection preferences, demand prediction, and systematic dispatch. Our study focused on the relationship between the trip-chain complexity environment and travel intention, combining the Theory of Planned Behavior (TPB) with travelers’ preferences to construct a bounded rationality theory. First, this study used K-means clustering to transform the characteristics of the travel trip chain into the complexity of the trip chain. Then, based on the partial least squares structural equation model (PLS-SEM) and the generalized ordered Logit model, a mixed-selection model was established. Finally, the travel intention of PLS-SEM was compared with the travel sharing rate of the generalized ordered Logit model to determine the trip-chain complexity effects for different public transport modes. The results showed that (1) the proposed model, which transformed travel-chain characteristics into travel-chain complexity using K-means clustering and adopted a bounded rationality perspective, had the best fit and was the most effective with comparison to the previous prediction approaches. (2) Compared with service quality, trip-chain complexity negatively affected the intention of using public transport in a wider range of indirect paths. Gender, vehicle ownership, and with children/without children had significant moderating effects on certain paths of the SEM. (3) The research results obtained by PLS-SEM indicated that when travelers were more willing to travel by subway, the subway travel sharing rate corresponding to the generalized ordered Logit model was only 21.25–43.49%. Similarly, the sharing rate of travel by bus was only 32–44% as travelers were more willing to travel by bus obtained from PLS-SEM. Therefore, it is necessary to combine the qualitative results of PLS-SEM with the quantitative results of generalized ordered Logit. Moreover, when service quality, preferences, and subjective norms were based on the mean value, with each increase in trip-chain complexity, the subway travel sharing rate was reduced by 3.89–8.30%, while the bus travel sharing rate was reduced by 4.63–6.03%.

## 1. Introduction

With urban traffic environment enhancement, as well as the accelerated pace of living, travelers increasingly need to visit multiple destinations in a single trip. It can involve a sequence of trip segments that begins with the “home” activity and continues until the traveler returns home [1]. Meanwhile, transportation departments have aimed to promote public transport by constructing various transportation infrastructures, especially a large amount of rail transit and integrated transportation hubs. However, different travel modes show great differences in travel chains, especially between multimode and single-mode travel chains [2]. Compared with cars, the lack of flexibility of public transport (e.g., long initial waits times, long driving distances) poses obstacles to travel via the travel chain, and such obstacles have only been intensified by the increasing complexity of travel chains [3,4,5]. Therefore, despite parking difficulties, rising oil prices, and a recognition of the importance of green travel, increasing numbers of travelers elect to have private cars. This aggravates related problems such as traffic congestion, pollution, and energy shortage.

Simply expanding the capacity of the transport infrastructure may not be an attractive congestion solution due to inefficient and costly capacity expansion [6,7]. The basic goal of transportation system planning is to solve the traffic balance problem. In recent years, with the rapid rise of multimodal transport planning research, researchers began to consider the capacity, allocation, pricing, and interaction of various modes [7,8,9]. However, the total demand has assumed to be fixed, and the supply–demand interaction in the multimodal transport system has been ignored. Including the demand transfer of multimodal transport in the balance model is another factor that has not been given due attention [10]. Furthermore, although it has been determined which urban public transport vehicles are acceptable and which are unacceptable based on strict economic and environmental requirements [11], h the above studies do not take into account the willingness of travelers to travel with different public transport tools in the context of the travel chain. Therefore, it is necessary to understand the travel mode choice of travelers in the context of the travel chain, and it is necessary to incorporate it into the balance model for better system planning.

Mobility as a service (MaaS) is a new transportation concept that uses intelligent digital technology to provide higher quality transportation services. Although there are some risks and challenges in the implementation of MaaS [12,13], relevant principles need to be followed [14]. MaaS provides us with a broad prospect of more effectively integrating, managing, and dispatching various transportation modes, and improves transportation efficiency compared with the single traffic mode adopted in the past to reduce congestion, save energy, and promote environmental protection [15,16,17]. Recently, the promotion of MaaS has had a positive impact on promoting public transport travel [18,19,20]. The MaaS platform especially promotes the travel of a complex travel chain with multiple transportation modes. In order to improve the quality of public transport service, it is necessary to accurately grasp the travel characteristics and selection preferences, and predict the traffic demand. The research aim is to explore the effects of the travel-chain complexity environment on public transport intention.

Considering the synergy between travel-chain complexity and service-quality perception, our study adopted a bounded rationality perspective and proposed a mixed-selection approach between partial least squares structural equation modeling (PLS-SEM) and ordered Logit econometric regression. Therefore, there are two main contributions of this study. The first is to combine TPB with preference to form the bounded rationality theory and introduce it into travel-chain research. The heterogeneity of the model is also considered. The second is to compare the results of SEM with those of ordered Logit, so as to obtain a combination of qualitative and quantitative research results. The specific research process is as follows. First, accurate measurement of the latent variables determined the validity of the model [21]. K-means clustering was used to transform travel-chain characteristics into travel-chain complexity, which solved the problem in which the potential variables of travel-chain complexity are difficult to obtain by a questionnaire. Then, through PLS-SEM, the interaction and heterogeneity of the travel-chain complexity environment and service quality in relation to public transport travel intention were investigated, and the PLS-SEM evaluation standard was improved. Finally, using an ordered Logit model, the relationship between travel intention and occurrence probability was further determined, and the marginal effect of the travel-chain complexity environment on different public transport modes was analyzed. This study is expected to support the mobility as a service (MaaS) transport environment. This can also help lay a foundation for balancing traffic based on MaaS.

The rest of this paper is organized as follows. Section 2 introduces the previous research and outcomes, and Section 3 presents the mixed-selection model framework. Section 4 describes the data collection and samples, while Section 5 presents the results. Section 6 analyzes the effect of heterogeneity on model functioning and compares the performance of different models. Section 7 summarizes the findings and discusses the prospects for future research.

## 2. Literature Review

As MaaS integrates, manages, and dispatches all kinds of transportation modes more effectively through digital technology to improve transportation efficiency [15,16], it is therefore necessary to study a series of travel mode choices in the travel process of travelers. Research on travel chains and travel modes can be divided into three main categories. The first category typically uses econometric regression models to study travel-chain and travel-mode decision-making processes. For example, Huang et al. (2021) and Qi et al. (2020) suggested that travel-chain decision making takes priority over travel-mode selection [22,23]; that is, travelers first organize the travel chain, which affects the choice of travel mode. Accordingly, the previous study focused on the effect of the travel-chain complexity environment on travel patterns. The second category of research studies the effect of travel chains on mode selection, which can be further divided into two subcategories. The first subcategory is research on travel behavior based on econometric regression models. Such work can provide insights into the types of travel chains and how they combine spatio-temporal relationships, travel purposes (commuting and non-commuting), climate, service quality, cost, or socio-demographic factors to affect travel-mode sharing and marginal effects [24,25,26,27,28,29]. There is currently relatively little evidence regarding user acceptance and the marginal effects of different public transport modes at different levels of travel-chain complexity. Researchers studying transportation behavior have come to realize that studying attitudes can improve our understanding of personal preferences and behaviors. Therefore, Structural Equation Modeling (SEM) has been adopted to study potential psychological variables such as attitude [21,30]; this is the second subcategory of research. Such work provides important insights into the effect of service quality on public transport use intention in the travel chain. For example, using SEM, Rahman et al. (2020) and Hadiuzzaman et al. (2019) found that the choice of travel mode was closely related to the comfort of the travel chain [5,24]. However, a key factor affecting service quality in the travel chain is transfer convenience [26,31]. Such variables have not been widely considered in research on the relationship between the travel chain and public transport travel intention. Meanwhile, SEM aims to model latent psychological variables. However, it is difficult to accurately determine the travel-chain complexity environment using questionnaire surveys, which gives rise to the problem of how to scientifically obtain the potential psychological variables of travel-chain complexity. Finally, the third category of research aims to identify the characteristics and functions of different traffic modes in the travel chain as well as travelers’ traffic-mode preferences [26,32]. There is, however, less empirical evidence from the perspective of bounded rationality. Considering the limited literature on that topic, the present study aimed to make a substantive contribution to transportation theory and methodology.

Transportation researchers often used the Theory of Planned Behavior (TPB) [33] to understand travel motivation and guide travelers to choose daily travel patterns. Yet, as noted by Herbert Simon in the 1950s, “economic man” actually has bounded rationality. Along these lines, Islam and Habib (2012) found that in actual travel processes, because of differences in travel habits and service preferences, travelers’ travel modes and travel decisions are often not the globally optimal choices [34]. Liao et al. (2020) found that travelers prefer different modes of transportation for different distances [26]. Moreover, different travelers have different preferences [24]. Few studies, however, have specifically investigated preferences in the travel chain from the perspective of bounded rationality [26]. This study, therefore, aimed to combine TPB with preference and introduce bounded rationality into travel-chain research, giving consideration to heterogeneity.

Regarding variables, the travel-chain complexity environment as considered in previous research usually concerns the type of travel chain; namely, the travel chain is considered a category variable. Studies often classify the travel chain as either “simple” (no intermediate stations) or “complex” (one or more intermediate stations) [22,28,35]. Econometric methods are often used to understand and quantify the causal relationship between travel complexity and mode selection. Some studies, meanwhile, have used more detailed classifications. Islam and Habib (2012) classified travel-chain types related to work into eight categories [34]. Schneider et al. (2020) specified simple (two trips), complex (three trips), and very complex (four or more trips) travel chains [2]. Currie and Delbosc (2011) regarded complexity as a continuous variable by considering the number of links in each chain [3]. Borriello et al. (2019) suggested that the incorrect measurement of latent variables results in the inaccurate expression of latent variable definitions and leads to poor model performance [21]. However, when conducting a survey, if you directly ask, “How do you feel about travel-chain complexity?” respondents might not understand what “travel-chain complexity” is and have difficulty giving an objective, accurate answer. Moreover, the latent psychological variables in SEM need to be continuous variables. It is important, therefore, to determine how to accurately obtain the latent psychological variables of travel-chain complexity. In this study, respondents were first asked about objective characteristics of the travel chain, such as the type of travel chain, the number of activity points in the travel chain, and the number of trips. Then, K-means clustering was used to map travel-chain complexity (very simple, simple, general, complex, and very complex) to ensure the effectiveness of the variables. In addition, since the key factor affecting service quality in the travel chain is transfer convenience [26,31], this study considered service quality related to transfer when examining the interaction between service quality and the travel-chain complexity environment.

With regard to research objects, aiming to reduce car travel in favor of public transport, most travel-chain research has focused on the choice between public transport and cars [4,5,30]. However, as the main travel modes in public transport, bus and rail transit have different operating characteristics, station density, line coverage, ticket prices, and service quality. Currie and Delbosc (2011) noted that the travel chain of rail transit in Melbourne, Australia, was more complex than that of bus transit [3]. Liao et al. (2020) found that in non-crowded short-distance travel, the marginal effect of buses in car time had a stronger negative preference effect than that of subways [26]. This raises the question of how much of a difference does it make for travelers to choose bus or rail transit for each additional unit of travel-chain complexity? That is, how does the marginal effect of travel-chain complexity on different public transport modes differ? Although this is an important consideration for balancing traffic modes in traffic-demand management, it is not investigated in the literature.

Although traffic-mode division and prediction are an important basis for traffic-structure optimization, few have studied the connection between travel willingness using PLS-SEM and the travel-mode sharing rate using a Logit model. It is easy to assume that in SEM, the highest willingness to use public transport represents a 100% chance of using it. In the complex travel-chain environment, however, willingness to use public transport does not necessarily correspond to a 100% probability of use. Individuals might be very willing to use public transport, but if there is no station nearby, it might be difficult for them to use it. It is also possible that private cars are needed because children need to be shuttled to school, and travel by public transport would be inconvenient in the context of travel chains.

Regarding research methods, studies from the past three decades have revealed that analyzing individual psychological factors and considering them in empirical models is important for capturing more behavioral changes. As a result, the hybrid discrete choice model (HDCM) has become increasingly popular in transportation research [21,36,37,38,39]. HDCM uses SEM to explore the composition of latent psychological variables and adds a classical selection model based on econometric regression. Although HDCM has prospects, it is rarely used in travel-chain research. In terms of travel-chain mode selection, Ye et al. (2018) established an SEM Logit model based on the intercity travel chain that introduced latent psychological variables [39]. The results showed that the model had a higher fitting degree and stronger interpretation ability, and it was more suitable for travelers’ actual decision-making processes. This method introduces SEM results into the Logit model as an effect function. However, it does not check whether all variables in SEM are suitable for the Logit model. The results of SEM were not compared with those of the Logit model. Therefore, the present study considered these issues.

The present study involved multiple latent variables, many intermediary relationships, and complex model estimation. PLS-SEM can not only allow the data variables to obey non-normal distribution but also overcome measurement error and obtain more accurate, reliable fitting results [40]. PLS-SEM was therefore considered a suitable method for this study. However, the evaluation indicators of PLS-SEM are different in different studies, and the precision of the results’ interpretation needs improvement. Referring to previous research, this study developed an improved model evaluation index system.

This study focused on the relationship between the trip chain complexity environment and travel intention, combining the Theory of Planned Behavior (TPB) with travelers’ preferences to construct bounded rationality theory and introduce it into travel-chain research.

Therefore, a mixed-selection model based on PLS-SEM and an ordered Logit model was developed to study the effect of travel-chain complexity on public transport travel intention. Figure 1 shows the logic diagram, while Table 1 presents the operational definitions of the variables. Assumptions are shown in Figure 1, such as H1 to H11 and their combination. The hypothesis includes the belief that the complexity of the travel chain and the quality of service have direct effects, intermediary effects, and regulatory effects on public transport travel intention from the perspective of bounded rationality. Additionally, from the perspective of bounded rationality, the marginal effect of travel-chain complexity on the travel intention of different traffic modes is different.

## 3. Modelling Approach

### 3.1. Transformation of Travel-Chain Characteristics Based on K-Means Clustering

Relying on the classification of travel chains using clustering [34,44], our study used K-means clustering to convert respondents’ answers of travel-chain characteristics into travel-chain complexity. As a result, travel-chain complexity was divided into five categories, in order to meet the requirements of the five-point Likert scale.

Based on the survey results, two cases required the K-means clustering transformation of the indicator variables. First, to meet SEM requirements, travel-chain types were converted into measurable latent psychological variables. Second, trip times (tripno) is a metric variable, but its maximum value exceeded five. Regarding the number of active points in the travel chain (activeno), since it is a continuous variable and the maximum value is five, it did not need to be clustered. It was directly divided into five categories according to the number of points: 1: very simple, 2: simple, 3: general, 4: complex, and 5: very complex.

(1)K-means clustering transformation for travel-chain type

Travel-chain type (chsize) was divided into seven types: HWH, HOH, HWOH, HOWH, HOWOH, HWOWH, and other; “H” stands for home, “W” stands for place of work; and “O” stands for other places. K-means clustering aims to judge the classification of cluster points as per the distance between the cluster points and the center point. Travel-chain types were grouped into five categories by K-means clustering, and the Euclidean distance between each point and the cluster center was calculated, as shown in Formula (1):(1)$$d_{kj}=\sqrt{\sum_{k=1}^{m}{\left(x_{k}-p_{j}\right)}^{2}}$$
where $x_{k}$ is the cluster center point where the travel-chain type is transformed into travel-chain complexity, k=1,2,3,4,5; and $p_{j}$ is the point to be clustered, j=1,2,3,……,390. When the $d_{kj}$ value is the smallest, the cluster corresponds to the cluster center, which is the set of the final similar trip chain type complexity. The five categories that correspond to travel-chain type are very simple, relatively simple, general, relatively complex, and very complex.

(2)K-means clustering transformation for trip times

Trip times were grouped into five categories by K-means clustering, and the Euclidean distance between each point and the cluster center was calculated, as shown in Formula (2):(2)$$C_{il}=\sqrt{\sum_{i=1}^{n}{\left(y_{i}-z_{l}\right)}^{2}}$$
where $y_{i}$ is the cluster center point where the number of trips is converted into the complexity of the number of trips, i=1,2,3,4,5; and $z_{l}$ is the point to be clustered, l=1,2,3,……,390. When the $C_{il}$ value is the smallest, the cluster corresponding to the cluster center is the set of the complexity of the final similar trip times. These five categories correspond to the complexity of travel times, i.e., very simple, relatively simple, general, relatively complex, and very complex.

### 3.2. PLS-SEM

The travel-chain complexity environment, preference, and service-quality perception are potential variables that cannot be directly measured. SEM enables including indirectly measured unobservable variables through indicator variables and calculating the measurement error of observed variables. Moreover, when the whole phenomenon occurs simultaneously and needs to be modeled, SEM is generally considered one of the best methods for measuring potential factors and evaluating the structural relationships between them [40]. Thus, SEM methods have been widely used to examine the overall framework of user behavior intention related to the use of public transport [45,46,47]. PLS-SEM is a type of SEM that mainly analyzes the principal component structure of variables and explains variations in dependent variables through theoretical models; it is used to develop theory through exploratory research [40].

The exogenous latent variables in this study were the travel-chain complexity environment and service quality. Preference, subjective norms, and public transport use intention were the endogenous latent variables. Based on travel-intention data from Shenzhen passengers, this study used SmartPLS 3.32 for the empirical analysis to verify the relationships between variables.

SEM comprises a measurement model and a structural model, as expressed by Equations (3)–(5). The measurement model was used to measure the relationships between latent variables and their corresponding observation variables; the structural model was used to detect causal relationships between the endogenous and exogenous variables. The model structure is as follows:(1)Basic equation of the measurement model
(3)$$x=\Lambda_{r}\xi +\delta ,$$
(4)$$y=\Lambda_{x}\eta +\delta .$$
where x is the q×1 vector of exogenous observation variables, $\Lambda_{x}$ is the q×n factor loading matrix of the influence of exogenous observation variables on exogenous potential variables, and δ is the q×1 vector of measurement error. Correspondingly, y is the p×1 vector of the endogenous observation variable, $\Lambda_{y}$ is the p×m factor loading matrix of the influence of the endogenous observation variable on the endogenous potential variable, and ε is the p×1 vector of the measurement error.

(2)Basic equation of the structural model:

(5)η=βη+Γξ+ζ
where η=m×1 is a vector composed of m endogenous latent variables, ξ=n×1 is a vector composed of n exogenous latent variables, and β=m×m is the coefficient matrix related to potential endogenous variables. Coefficient matrix Γ=m×n is related to potential exogenous variables. The residual vector ξ=m×1 is associated with the endogenous variable.

Referring to [40], PLS-SEM analysis was conducted by two stages. First, the measurement model was evaluated to determine whether it had sufficient reliability and validity. Second, the structural model was evaluated to understand the establishment of the research hypothesis.

### 3.3. Principle of Ordered Logit

Econometric regression analysis, including travel probability (generally expressed by ordered values such as “very small probability”, “relatively small probability”, “general,” “relatively large probability,” and “very large probability”), is often based on ordered Logit (also known as ranking models) [48,49]. To analyze the influence of preference data, service quality, subjective norms, and the travel-chain complexity environment on the travel mode sharing rate, this study estimated a Logit ranking model for bus and rail transit. The marginal contribution and probability prediction methods of postestimation were used to further analyze the effect of the travel-chain complexity environment on buses and subways. SPSS 22.0 is the general name of a series of software products and related services for statistical analysis, data mining, predictive analysis and decision support tasks launched by IBM. It was used to model the ordered Logit model, and Maximum Likelihood Estimation (MLE) was used for estimation.

According to previous analysis, it is feasible to use ordered Logit regression to estimate discrete multiclass selection data. In our study, travel probability measurement involved discrete data characterized by classification, not continuous statistical data; specifically, very small probability = 1, relatively small probability = 2, generally = 3, relatively large probability = 4, and very large probability = 5. In the ordered Logit model, the ordered response taken on (1, 2,…5) can be expressed by Formula (6).

Specifically, in our study, the ordered response with values $Y_{i}$ taken on (1, 2,…5) was expressed. The ordered Logit model of bus-trip probability is expressed by Formula (6):(6)$$Y_{i}^{*}=\alpha X_{i}+\varepsilon ,i=1,2,\ldots ,5,$$
where *ε* is an independent and identically distributed random variable, subject to logistic distribution, i.e., $\varepsilon_{i}|x\sim Logit(0,1)$; $Y_{i}^{*}$ is a latent variable; *α* is the parameter to be evaluated; and the explanatory variable (independent variable) is represented by matrix x. The relationship between the unobservable latent variable $Y^{*}$ and the observable variable y is expressed by Formula (7):(7)$$Y_{i}=\left\{ \begin{array}{l} 1,\;Y_{i}^{*}\leq \beta_{1}\\ 2,\;\beta_{1}<Y_{i}^{*}\leq \beta_{2}\\ \vdots ,\;\vdots \\ 5\;Y_{i}^{*}>\beta_{4} \end{array}\right.,$$
where $\beta_{1}$, $\beta_{2}$, $\beta_{3}$ and $\beta_{4}$ are the parameters to be estimated, called “tangent points”. In the ordered Logit model, bus-trip probability *y* = 1, 2,…5, as shown in Formula (8):(8)$$P(Y_{i}=1|X_{i})=P(\alpha X_{i}+\varepsilon_{i}\leq \beta_{i}|X_{i})=\Phi (\beta_{i}-\alpha X_{i})\;P(Y_{i}=2|X_{i})=P(\beta_{1}<\alpha X_{i}+\varepsilon_{i}\leq \beta_{2}|X_{i})=\Phi (\beta_{2}-\alpha X_{i})-\Phi (\beta_{1}-\alpha X_{i})\;P(Y_{i}=3|X_{i})=P(\beta_{2}<\alpha X_{i}+\varepsilon_{i}\leq \beta_{3}|X_{i})=\Phi (\beta_{3}-\alpha X_{i})-\Phi (\beta_{2}-\alpha X_{i})\;P(Y_{i}=4|X_{i})=P(\beta_{3}<\alpha X_{i}+\varepsilon_{i}\leq \beta_{4}|X_{i})=\Phi (\beta_{4}-\alpha X_{i})-\Phi (\beta_{3}-\alpha X_{i})\;P(Y_{i}=5|X_{i})=P(\alpha X_{i}+\varepsilon_{i}>\beta_{4}|X_{i})=1-\Phi (\beta_{4}-\alpha X_{i}).$$

The marginal effect of bus-trip probability is calculated as shown in Formula (9):(9)$$\begin{array}{r} \partial P(Y_{i}=1)/\partial X_{i}=-\varphi (\beta_{1}-\alpha X_{i})\alpha \\ \vdots \;\vdots \;\vdots \;\\ \partial P(Y_{i}=5)/\partial X_{i}=\varphi (\beta_{4}-\alpha X_{i})\alpha \end{array}.$$

In Formulas (8) and (9), Φ(•) is the cumulative distribution function of Logit distribution, so it is an ordered Logit model; and φ(•) is the probability density function.

MLE was adopted for model parameter estimation; the MLE function is constructed as follows:(10)$$L_{i}(\alpha )={\prod_{j=1}^{5}\left[P(Y_{i}=j|X)\right]}^{1(Y_{i}=j)}.$$

Then, the log-likelihood function is $\ln L_{i}(\alpha )\sum_{j=1}^{5}1(Y_{i}=j)P(Y_{i}|X)$, where 1(•) is an indicator function. To maximize the sample logarithm likelihood function, the coefficient estimate ${\hat{\alpha }}_{MLE}$ can be obtained by solving Formula (10), as shown in Formula (11):(11)$$\frac{\partial \ln L_{i}(\hat{\alpha })}{\partial \hat{\alpha }}=\sum_{j=1}^{5}1(Y_{i}=j)\frac{\partial P(Y_{i}=j|X)}{\partial \hat{\alpha }}=0.$$

Rail transit travel probability is consistent with the basic principle of bus ordered Logit, except that there was no sample with “very small probability” in the questionnaire. Therefore, travel probability *y* = 2, 3,…5.

## 4. Data Collection and Sample Description

### 4.1. Data Collection

This study used the questionnaire survey to collect data. In 2018, travelers using public transport at bus stops, subway stations, shopping malls, or transport hubs in Shenzhen, China, were selected as the sample objects. Ten investigators were trained to conduct a 14-day investigation. A total of 450 questionnaires were collected. After removing invalid samples, including missing, contradictory, or incomplete answers, the number of valid questionnaires was 390 (effective response rate: 86.7%).

The questionnaire was divided into two parts. The first part covered demographic information, including age, gender, annual income, education level, with children/ without children, and vehicle ownership. The second part included the variable information, which was used to explore the decision-making process between travel-chain characteristics and public transport use intention. The variable information was divided into five dimensions (22 measurement indicators): travel-chain characteristics, service quality, subjective norms, preferences, and use intention. The questionnaire used five-point Likert scales. For example, service quality ranged from 1 (very poor) to 5 (very good). The questionnaire design was revised by the doctoral supervisor in the field of transportation planning and management and two senior experts in the field of statistics and economic management.

### 4.2. Sample Description

Among the respondents, 220 (56.4%) were male, and 230 (59%) were below 30 years. Regarding annual income, 209 (53.6%) made less than RMB 80000. For education level, 204 (52.3%) were below the junior college level. A total of 41.5% had children, and 67.2% had cars.

Figure 2a depicts the heterogeneous samples. In this study, heterogeneous sample analysis was conducted according to gender, with/without a car, and with/without children. Figure 2b shows the effect of travel-chain complexity on public transport travel intention with/without children. It shows that more than half of the respondents with children needed to travel in a very simple travel chain. Most were generally willing or relatively willing to travel by public transport, while a few were very reluctant to use public transport, which was slightly more than the number of travelers without children. When the travel chain was complex or very complex, travelers with children were not very willing to use public transport. Figure 2c shows respondents’ perceptions of the service quality of different public transport types. The service-quality perception of subways (e.g., comfort, safety, convenience, speed, transfer convenience, and the preferences of relatives, friends, and themselves) was generally higher than that of buses, and the difference between speed and safety was relatively large. Figure 2d shows the change in public transport travel willingness with the increase in travel-chain complexity. Among them, latrs1 represents whether future trips will be dominated by buses, and latrs2 indicates whether they will be dominated by the subway; latwt1 represents willingness to travel by bus in the future, and latwt2 the willingness to use the subway in the future. It can be seen that for both bus and subway, with the complexity of the travel chain increasing from level 1 to level 5, the willingness to travel gradually decreases from 4.450 (red) to 2.950 (blue). The change direction on the figure reflects this rule. Additionally, compared with buses, respondents preferred to use the subway when the travel chain was complicated. Although the above phenomena are clearly observed in the figure, the underlying mechanisms and the marginal effect of travel-chain complexity on different public transport modes are not clear. This study further investigated this issue so that effective measures for promoting public transport could be identified.

## 5. Data Analysis and Model Evaluation

### 5.1. Travel-Chain Characteristics Transformed into Travel-Chain Complexity

Table 2 and Table 3 show the clustering results. The K-means clustering results for travel-chain types and trip times met the significance requirements (*p* < 0.05).

### 5.2. Measurement Model Analysis and Evaluation

Because the model was constructed by using reflective indicators, question reliability, latent variable reliability, convergent validity, and discriminant validity needed to be reported in the measurement model [40].

#### 5.2.1. Factor Loading and Significance Test

Question reliability is the square of the factor loading, and the meaning is how much the latent variable explains question variance. Fornell and Larcker (1981) suggested 0.5 or above to be ideal [50]. Therefore, question reliability is based on a factor loading greater than 0.7, which is significant at the level of 95%. However, when the factor loading is between 0.4 and 0.7, it only needs to be deleted when deleting the factor loading will increase composite reliability and average variance extracted (AVE) to the recommended threshold [40]. PLS-SEM parameter estimates are standardized values, and significance estimation needed to be conducted by bootstrapping to obtain the standard error of the normalization coefficient for significance calculation.

Table 4 shows that the factor loadings (original sample) of all questions measuring latent variables were between 0.565 and 0.838, reaching a significance level of 95%; |t| > 1.96, and *p* < 0.05, indicating that all questions had reliability.

#### 5.2.2. Reliability, Convergence, and Discriminant Validity

Cronbach’s α is the traditional indicator used to evaluate the reliability of latent variables. It assumes that all indicators are equally reliable (i.e., the factor loading of each indicator on the latent variable is equal), which is different from the assumption of PLS-SEM. In addition, Cronbach’s α is sensitive to the number of scales and tends to underestimate internal consistency. Therefore, for PLS-SEM, a more appropriate indicator is composite reliability, which takes into account the difference in the composite amount of indicator factors. Composite reliability is better between 0.7 and 0.9 [40]; Chin (1998a) suggested that 0.8 or above is ideal [51]. Cronbach’s α will be slightly lower than the composite reliability [52]. Table 5 shows that the Cronbach’s reliability of potential variables α is 0.564–0.824, and composite reliability is greater than 0.8. Therefore, the latent variables in this study had good reliability.

Convergent validity refers to the degree of positive correlation between the measurement indicators to which the latent variable belongs. Testing convergent validity requires considering the factor loading and AVE of the index. The recommended AVE value is 0.36 or above [53]. AVE > 0.5 indicates sufficient convergent validity [50]. Table 5 shows that the AVE of most latent variables was 0.560–0.697, indicating sufficient convergent validity.

Discriminant validity pertains to the degree of difference between latent variables. There are two methods in PLS-SEM for testing discriminant validity. First, Fornell and Larcker (1981) suggested that the open root value of the latent variable’s AVE should be compared with the Pearson’s correlation of the other latent variables [50]. If the difference is large, it indicates there is discriminant validity. In Table 5, the root opening number of the potential variable’s AVE is greater than that of other potential variables. Therefore, the discriminant validity between latent variables is good.

Cross-loading is another method for testing discriminant validity [40]. The factor loading of the latent variable itself should be greater than the loading of the latent variable and other latent variables (cross-loading). Testing revealed that the factor loading (bold) was greater than the cross-loading, indicating that the latent variables in the model had good discriminant validity (for reasons of length, this is not repeated; see Table A1 in the Appendix A for details).

### 5.3. Structural Model Analysis and Evaluation

We can see from the above that all potential variables had suitable reliability, convergent validity, and discriminant validity. Regarding the structural model, previous studies using PLS-SEM often only used structural model coefficients and determination coefficients as the verification standard. This can lead to insufficiently complete model evaluation. To more comprehensively evaluate the structural model, based on the literature, this study established more complete indicators for structural model evaluation, including collinearity (variance inflation factor; VIF), the path coefficient of the structural model, the determination coefficient, Stone–Geisser’s Q2 value, and goodness of fit (GOF). These indicators were used to comprehensively verify the model, as shown in Table A1 in the Appendix A.

SmartPLS 3.29 was used to verify the structural model and analyze the causal relationships between latent variables. The bootstrapping method was used for repeated sampling analysis. Bootstrapping takes the existing sample number as the repeated sample of the mother to rebuild a new sample that can represent the distribution of the mother sample. Its advantage is that it can be inferred and analyzed without strict hypothesis testing on the distribution characteristics; this is because the source distribution used is the distribution from real data. Figure 3 shows the causality of the PLS-SEM study pattern.

#### 5.3.1. Collinearity (VIF) Evaluation

Unlike reactivity indicators, which can be replaced by each other, formative indicators should not be highly correlated. The correlation between these indicators is also called collinearity. In this study, each index variable was reactive to the latent variable while the latent variables of the model were formative. Therefore, only the collinearity of the latent variables of the model were evaluated. Table A1 in the Appendix A and Table 6 show that all VIF values satisfied 0.2 < VIF < 5; thus, there was no collinearity among the internal latent variables.

#### 5.3.2. Overall Structural Model Evaluation

One criterion for evaluating PLS-SEM is the interpretable variance ($R^{2}$) of the endogenous latent variable, which is a standardized estimate; the closer to 1, the better the independent variable selection. It is not easy, however, to propose a generally acceptable $R^{2}$, because it involves the complexity of models and subject differences. It is noted that an $R^{2}$ value of 0.20 is high in customer behavior research [40]. Table A2 in the Appendix A and Table 7 show that when $R^{2}$ ≤ 0.02, the path relationship is weak. When 0.02 < $R^{2}$ ≤ 0.13, the path relationship is medium, and when 0.13 < $R^{2}$ ≤ 0.26, the path relationship is strong. In this study, most were between 0.319 and 0.493, showing strong interpretation ability and effective model parameter calibration.

The second criterion for evaluating PLS-SEM is the value of Stone–Geisser’s $Q^{2}$. $R^{2}$ calculates the residual for the samples participating in the fitting to reflect the “fitting ability”; the $Q^{2}$ residual is calculated for samples not participating in the fitting in the cross-validation, which reflects “prediction ability.” The result must be greater than 0; the greater the prediction correlation, the stronger the prediction correlation. It can be obtained using the blind solution. As shown in Table A2 of the Appendix A and Table 7, cross-validated redundancy was used to calculate the values. All values were between 0.030 and 0.254, which means the requirements were met.

The third criterion for evaluating PLS-SEM is GOF, which is used to evaluate the overall fitness of the model. Formulas (12) and (13) show the calculation method, where AVE is the AVE quantity, representing the quality of the latent variables, and n is the quantity of latent variables. $R^{2}$ (coefficient of determination) is the coefficient of determination, which represents the quality of the model structure, and m is the number of endogenous latent variables. As shown in Table A2 in the Appendix A, GOF = 0.947 > 0.360, which is large. Thus, the overall GOF of the model was good.
(12)$$GOF=\sqrt{\mathrm{Commonality}\times \mathrm{Interpretable}\;\mathrm{variance}}=\sqrt{\frac{\sum_{i=1}^{n}AVE_{i}}{n}+\frac{\sum_{m=1}^{m}R_{n}{}^{2}}{m}},$$
(13)$$\begin{array}{l} GOF=\sqrt{\frac{0.645+0.697+0.560+0.593+0.414}{5}+\frac{0.404+0.319+0.458+0.076}{4}}\\ =0.947. \end{array}$$

Hence, the structural model had good fitting and prediction ability.

#### 5.3.3. Path Analysis

Figure 3 and Table 7 show the calculation results for the model optimization. The path coefficient check includes whether the positive and negative directions conform to the hypothesis and strength (coefficient size) detection. Chin (1998a) suggested that the path coefficients should be at least greater than |0.2|, and the significance of the path coefficient should be under the 95% confidence level [51]. Table 7 shows that most of the direct effects of the path coefficients are between |−0.2768| and |0.523|, which conforms to [51]. A small part does not meet the recommended standard. However, the model needs to analyze not only the direct effects but also the intermediary effects through other potential variables. This will be reported below.

The significance of the coefficient ultimately depends on the standard error obtained from bootstrapping [40]. Bootstrapping was used to calculate the statistics of each path coefficient and test the significance level of the path coefficient estimation (two-tailed test). If 2.58 > T > 1.96, the path coefficient was considered significant at the level of 0.05. If 3.29 > T > 2.58, the path coefficient was considered significant at the level of 0.01. If T > 3.29, the path coefficient was considered significant at the level of 0.001. The SEM statistics in the bootstrapping test showed that the direct effects of all path coefficients had high statistics, indicating that each path coefficient passed the test of the corresponding significance level, further indicating that the stability of the model structure was good after 5000-sample repeated sampling (Table 7). At the same time, the probability of rejecting the null hypothesis was less than 0.001, further showing that the significance level of the path coefficient was high. It can be seen in Table 7 that the sign of the path coefficient also conformed to the research hypothesis, indicating that all hypotheses were valid.

In summary, first, the results verified that travel-chain complexity had a negative and significant effect on service quality, subjective norms, and preferences. Since travel-chain length runs counter to the convenient, economical, and rapid evaluation criteria of public transport service quality, the longer the travel chain, the lower the service quality perceived by users. The results also showed that the longer the travel chain, the less friends and relatives supported it, and the greater the number of transfer nodes, the less likely users were to prefer public transport.

Second, service quality positively affected subjective norms and preferences. When the safety, speed, convenience, and comfort of public transport were higher, the attitude of family and friends toward public transport was more positive. Family members usually hope they will not travel too much. Meanwhile, travelers formed preferences for certain modes of transit because of the fast and comfortable experiences.

Third, subjective norms and preferences had a positive and significant effect on the intention to use public transport. Since preferences reflect individuals’ most efficient choices, and subjective norms are the attitudes of important relatives and friends, both were found to have significant effects on the intention to elect public transport. This is consistent with [54].

#### 5.3.4. Intermediary Effects

In PLS-SEM, if the effect of a latent variable can only occur through another latent variable, it is called an indirect or intermediary effect. Mediating effect significance was estimated using bootstrapping with 5000 repetitions. Table 8 lists the paths of the intermediary effects. The results showed that all intermediary effects were valid.

The intermediary effect analysis first established indirect effects a×b, which tested whether these indirect effects were significant. Then, the difference between the total effect and direct effect was tested. Based on [55], Nitzl et al. (2016) proposed a way to analyze the mediation effect in PLS-SEM [56], as shown in Formula (14):(14)$$\mathrm{Total}\;\mathrm{effect}\;(c)=\mathrm{indirect}\;\mathrm{effect}\;(a\times b)+\mathrm{direct}\;\mathrm{effect}\;(c^{\prime }),$$

The effects of service quality and the travel-chain complexity environment on public transport use intention are both achieved through indirect effects. The total effect of service quality on public transport travel intention is as shown in Formula (15) [40,56]:(15)H3×H8+H5×H7+H3×H6×H7=0.194+0.068+0.096=0.358.

Based on Formula (15), the comprehensive path coefficient of the effect of service quality on public transport use intention was 0.358, indicating a positive effect.

Based on the travel-chain complexity environment, either through the mediation effects of subjective norms and preferences or through the remote mediation of service quality affecting preferences and subjective norms, the mediation effects were negative. The total effect of travel-chain complexity on public transport travel intention is shown in Formula (16):(16)$$\begin{array}{l} H4\times H7+H1\times H3\times H8+H2\times H8+H1\times H5\times H7+H2\times H6\times H7+\\ H1\times H3\times H6\times H7=(-0.045)+(-0.054)+(-0.041)+(-0.019)+\\ (-0.020)+(-0.026)=-0.205 \end{array},$$

First, according to the calculation results of Formula (16), the comprehensive path coefficient of the effects of the travel-chain complexity environment on public transport travel intention met Chin’s (1998a) requirement [51]—namely, it should be greater than |0.2|. Thus, the travel-chain complexity environment had a significant negative effect on public transport travel intention. Second, if the total effect was not considered, the direct effect of travel-chain complexity or of service quality on public transport travel intention was 0, which falls short of Chin’s (1998a) standard [51]. It tends to result in neglecting the effect of travel-chain complexity or service quality on public transport travel intention, thus losing important research results. Finally, as shown in Table 9 and the results from Formulas (15) and (16), service quality had a greater effect on public transport use intention than the travel-chain complexity environment through the total effects. However, compared with positive service quality, the travel-chain complexity environment had negative effects on public transport use intention in a wider range of paths.

### 5.4. Ordered Logit

First, Table 10 and Table 11 show that none of the respondents were “very unwilling” to take the subway. Therefore, willingness to travel by subway was divided into four levels (relatively unwilling, general, relatively willing, very willing), and bus willingness was divided into five levels (very unwilling, relatively unwilling, general, relatively willing, very willing). According to the results of the ordered Logit model, the classification standard was significant.

Second, not all variables suitable for SEM were suitable for ordered Logit regression. The variables that could be used for ordered Logit modeling were obtained through experiments, and willingness to take buses and subways was modeled, as shown in Table 10 and Table 11 and Formulas (17) and (18):(17)$$\begin{array}{l} \ln(\frac{y_{bus}}{y_{bus}-1})=(-0.250)\times ChSize1+0.307\times Q1vbus\_mc+0.377\times Q1trfbus\_mc+\\ 0.296\times eval1\_mc+0.483\times eval6\_mc+1.199\times habby1\_mc \end{array},$$
(18)$$\begin{array}{l} \ln(\frac{y_{sub}}{y_{sub}-1})=(-0.262)\times activeNo+0.526\times Q1ssub\_mc+0.470\times Q1stsub\_mc+\\ 1.228\times eval7\_mc+1.073\times habby2\_mc \end{array},$$
where $y_{bus}$ is willingness to choose bus travel, and $y_{sub}$ is willingness to choose subway travel.

To clearly explain the effect of each unit change in travel-chain complexity on bus or subway travel willingness, independent variables other than travel-chain complexity were centered. Taking Q1vbus as an example, the coefficient value of each variable was not affected before and after centralization; the centering process is shown in Formula (19):(19)Q1trfbus_mc=Q1trfbus−Q1trfbus_mean,
where Q1trfbus_mc is the value after centralization and Q1trfbus_mean is the mean value of Q1trfbus.

As per Formula (17), travel-chain types in travel-chain complexity, bus convenience, bus transfer convenience, influence of family or friends, and preference had significant effects on the probability of choosing the bus. Among them, preference had the greatest effect, while the negative effect of travel-chain type cannot be ignored.

As Formula (18) shows, the number of travel destinations in travel-chain complexity, subway safety, subway speed, influence of family or friends, and preference had a significant effect on the probability of choosing the subway. Among them, friends and preferences had great effects, while the negative effect of the number of destinations in the travel chain cannot be ignored.

## 6. Comparative Analysis of Model Performance

### 6.1. Multigroup Comparison

Multigroup analysis was conducted for gender, with children or without children, and vehicle ownership to understand whether the influence coefficient of different population variables on each path was different. Figure 2a depicts the heterogeneity analysis sample. Men accounted for 56.4%, and women accounted for 43.6%; 41.8% had children, and 58.2% did not; and 67.2% had a private car, and 32.8% had no cars.

As shown in Table 12, under the comparison of gender groups, there were significant differences in the travel-chain complexity → preference paths (Δ Path (male–female) = −0.200, *p* = 0.028), indicating that in travel-chain complexity → preference paths, the negative effect of women on the path was lower than that of men. Namely, the more complex the travel chain, the more reluctant men were to use public transport.

As shown in Table 13, there was a significant difference in one path between with/without a vehicle—preference → subjective specification (Δ Path (with car–without car) = −0.240, *p* = 0.006)—indicating that those without cars were more likely to be affected by the subjective norms of public transport travel after they had a preference for public transport services.

As shown in Table 14, in the with/without children comparison, there was a significant difference in service quality → preference (Δ Path (with children–without children) = 0.222, *p* = 0.007), indicating that the effect of public transport service quality on preference was higher for those with children than for those without. This suggests that once service quality is improved, travelers with children are more likely to form a preference for public transport travel; this is similar to [57].

### 6.2. Comparative Analysis of Structural Models

PLS-SEM without K-means clustering is named as Model 1, as shown in Figure 4a. The model designed from the perspective of TPB is named as Model 2, as shown in Figure 4b. The PLS-SEM model using K-means clustering and designed from the perspective of bounded rationality (i.e., the model proposed in this study) is named as Model 3 shown in Figure 3.

The reliability and validity of Models 1, 2, and 3 were verified (in Table A3 in the Appendix A), laying a foundation for the further comparison of the models.

First, Table 15 shows that compared with Model 1, the prediction ability of Model 3 improved by 7.64% and GOF improved by 4.76%. This indicates that using K-means clustering to transform travel-chain characteristics into travel-chain complexity improved the performance of the model.

Second, compared with Model 2, the collinearity of Model 3 was optimized by 6.62%, prediction ability within the sample was improved by 6.52%, prediction ability outside the sample was improved by 2%, and GOF was improved by 4.99%. This shows that the overall performance of the model was enhanced after the bounded rationality perspective was added.

Third, “travel-chain complexity → travel intention,” the path coefficients of Model 1 and Model 2, did not meet the requirements of the standard. After the bounded rationality perspective was added to Model 3, the path coefficient increased by 22.29%, and the absolute value of the effect of travel-chain complexity on travel intention was greater than 0.2, achieving the path coefficient specified by [51] (i.e., the effect is significant when the absolute value is greater than 0.2). This is important, otherwise, travel-chain complexity, which is an important factor affecting travel intention, would be missed, resulting in a significant deviation in the results. Therefore, Model 3 reflects the important effect of travel-chain complexity on travel intention.

In summary, Model 3 with K-means clustering transformation and a bounded rationality perspective had the best fit and effectiveness.

### 6.3. Comparison of Regulation Effect of PLS-SEM before and after K-Means Clustering

As shown in Table 16, in the regulation effect analysis, although the regulation effect of “gender” on “travel-chain characteristics → preference” existed in Model 1, the effect was too small, and the intensity was only |−0.026 *|. After transformation, the regulation effect of Model 2 increased by 669.23%, and the intensity reached |−0.200 *|, just reaching the standard of [51]. Otherwise, the regulation effect of gender on travel-chain complexity → preference would be ignored, leading to the omission of important results.

### 6.4. Comparison of the Marginal Effects of Travel-Chain Complexity on Buses and Subways

Marginal contribution is used to analyze the effect on the probability that an individual belongs to a specific category when the independent variable changes while other variables remain unchanged [58]. To understand the change in the public transport travel probability category with a change in the independent variable value, we studied the marginal contribution of travel-chain complexity to the public transport travel probability category when taking the mean value of other independent variables [59].

As shown in Table 17 and Table 18, the coefficients of ordered Logit regression could not represent the marginal effect. This is because in a non-linear model, the marginal effect is not a constant but varies with changes in the explanatory variables. Therefore, the marginal effects need to be calculated one by one, as shown in Formulas (20)–(25). In addition, as shown in Figure 2c, the average values of the latent psychological variables regarding service quality, preference, and subjective norms of travelers, whether they take the subway or bus, all exceeded three (i.e., in the range of “general” to “relatively satisfactory”). This study considered the effect of travel-chain complexity on bus or subway travel willingness level under the premise that the latent psychological variables of service quality, preference, and subjective norm are all within the mean value.

Regarding buses, the formula was changed to leave only travel-chain complexity, as shown in Formula (20) and Formula (21):(20)$$\ln(\frac{y_{bus\_m}}{1-y_{bus\_m}})=(-0.250)\times ChSize1,$$
(21)$$y_{bus\_m}=e^{(-0.250)\times ChSize1}.$$

When the value of the independent variable ChSize1 is $x_{k}=1,2,3,4,5$, the marginal effect calculation Formula (22) of the corresponding bus travel probability is as follows:(22)$$\frac{\partial y_{bus}}{ChSize1}=\frac{\partial e^{(-0.250)\times ChSize1}}{ChSize1}=e^{-0.250},$$

Regarding subways, Formulas (20) and (21) were reformulated to handle travel-chain complexity, as Formulas (23) and (24):(23)$$y_{sub\_m}=e^{(-0.262)\times activNo},$$
(24)$$\ln(\frac{y_{sub\_m}}{1-y_{sub\_m}})=(-0.262)\times activeNo,$$

When the value of the independent variable activeNo is $x_{j}=1,2,3,4,5$, the marginal effect of the corresponding subway travel probability is calculated as shown in Formula (25):(25)$$\frac{\partial y_{sub}}{\partial activNo}=\frac{\partial e^{(-0.262)\times activNo}}{\partial activNo}=e^{-0.262}.$$
where $y_{bus\_m}$ is bus travel probability when variables other than travel-chain complexity are the average; $y_{sub\_m}$ is subway travel probability when variables other than travel-chain complexity are the mean value.

Odds represents the odds of taking a certain mode of transportation. If *odds* > 1, the probability of occurrence of the event is high; if *odds* = 1, the possibility of this event occurring is the same as that of not occurring; and if *odds* < 1, the probability of occurrence of the event is high. First, as shown in Table 17 and Table 18, and Figure 5a, as travel-chain complexity increases, the probability of travel by bus and subway decreases, and the subway decreases a little faster than the bus. Second, Table 17 and Table 18, and Figure 5b show that as travel-chain complexity increases, the change rate of the reduction in subway and bus travel probability gradually increases, and the change rate of subway odds is greater than that of bus odds. When the number of subway travel destinations increases from one to two, subway travel probability is reduced by 8.30%; when the number reaches three, subway travel probability is reduced by 3.89%. On that basis, if one travel destination is added, subway travel probability will be reduced by 5.34%. By comparison, the effect of the complexity of the bus travel chain is relatively balanced. When each unit is added, bus travel willingness decreases by 4.63–6.03%. Moreover, when travel-chain complexity is at level 2 and level 3, the change rate of subway and bus travel willingness is consistent. However, when travel-chain complexity is at level 4, the change rate of subway travel willingness is greater than that of the bus. At level 5, it overlaps again. This differs from [26], who found that in non-crowded short-distance travel, the marginal effect of time on the bus had a stronger negative preference than that on the subway. Our study used travel-chain complexity instead of in-vehicle time, and a more comprehensive result was obtained. The reason for the difference could be that when travel-chain complexity is at level 4, it is no longer a short trip but a long trip. According to the demand curve in microeconomics, price is inversely proportional to the quantity of goods purchased. As distance increases, subway fares increase more than bus fares. Then, the marginal effect of the travel-chain complexity of the subway will have a stronger negative preference effect than that of the bus. However, when distance increases further and travel-chain complexity is at level 5, since the comfort and speed of the subway are better than those of the bus, the change rate of bus or subway travel willingness will coincide.

Additionally, as shown in Figure 5b, for each unit increase in travel-chain complexity, the change rate of subway travel probability was slightly faster than that of the bus, but the respective values remained constant. The decrease rate of subway *odds* was 0.23, while the change rate of bus *odds* was 0.22. This is consistent with the basic preference for the bus being slightly higher than that for the subway, in accordance with Liao et al. (2020), and the basic preference for the subway is the lowest for long-distance travel.

Finally, Table 17 and Table 18 raise some noteworthy issues. First, not all variables suitable for PLS-SEM are suitable for an ordered Logit model. Second, when travel-chain complexity increased from 1 to 5, subway travel willingness was 4 according to PLS-SEM. Namely, travelers prefer to travel by subway. However, according to the ordered Logit model, subway travel probability was 21.25–43.49% (i.e., less than 50%). When travel-chain complexity was at levels 1, 2, and 3, bus travel willingness was 4; namely, willingness was higher, with bus travel probability at 32–44%. When travel-chain complexity was at levels 4 and 5, bus travel willingness was 3 (i.e., “generally willing”), and bus travel probability was 22–27%. This indicates that there were differences between travel intention level in PLS-SEM and share rate under ordered Logit. Under the same travel intention level, the subway share rate changed within a larger range than that of the bus. As shown in Figure 5b, the change rate in subway odds was greater than that for bus odds. In conclusion, as travel-chain complexity increased, the uncertainty of taking the subway was greater than that for taking the bus.

## 7. Conclusions

Based on a survey of Shenzhen residents in 2018, this study established a mixed-selection model to study the travel intention of public transport in the complex travel chain environment and obtained the following conclusions:

(1) The bounded rationality perspective and the potential variable of travel-chain complexity are of great significance to improve the public transport environment. In this study, travel-chain characteristics were transformed into the latent psychological variables of travel-chain complexity by K-means clustering, and the PLS-SEM obtained from the perspective of bounded rationality had the best fit and validity. The results showed that although travel-chain complexity and service quality had no direct effect on public transport travel intention, they had important indirect total effects. Compared with service quality, travel-chain complexity negatively affected the intention to use public transport in a wider range of indirect paths. In order to promote public transport travel as a feasible policy, it is necessary to consider not only the improvement in service quality but also the impact of travel-chain complexity on public transport travel willingness.

(2) Gender, vehicle ownership, and with children/without children had significant moderating effects on certain paths of the SEM. Namely, the travel-chain complexity of different groups had different effects on public transport travel intention. For example, the negative influence coefficient of women’s preference for travel-chain complexity was lower than that of men, which could be because men are impatient with long travel chains. The influence of preference on subjective norms was significantly higher for those without cars than for those with cars. A possible reason is that travelers without cars take public transport more frequently than those with cars. The preference for public transport travel will have a greater effect on subjective norms. Therefore, the public transport willingness for this part of travelers can be strengthened by increasing the propagation of green travel. Furthermore, having children had a greater effect on the route of public transport service quality to preference than for those without children. This indicates that respondents with children were more willing to take public transport as long as the quality of public transport service was improved.

(3) The travel-chain complexity environment had different marginal effects on different public transport modes. When service quality, preferences, and subjective norms were at the average level, subway travel probability was reduced by 8.30–3.89% and bus travel probability was reduced by 4.63–6.03% with each level of increase in travel-chain complexity. When travel-chain complexity was level 2 and level 3, the change rate of subway and bus travel probability was consistent. However, when travel-chain complexity was at level 4, the change rate of subway travel willingness was greater than that of the bus. At level 5, it overlapped again. In addition, the meaning of public transport travel intention was different under PLS-SEM than under the ordered Logit model. When subway travel intention obtained by PLS-SEM was “relatively willing,” the travel probability corresponding to the ordered Logit model was only 21.25–43.49%, and bus travel probability was 32–44%. That is, as the complexity of the travel chain increases, the uncertainty of taking the subway is greater than taking the bus. It is necessary to combine the qualitative results of PLS-SEM with the quantitative results of generalized ordered Logit to obtain more comprehensive and objective results.

Moreover, the MaaS platform uses the collected data on travelers’ travel patterns and preferences, optimizes the network for operators by calibrating supply and demand, and provides real benefits for travelers in the form of improving travel choices, saving time, reducing costs, and providing a better service experience [60]. The findings of our study can support personalized marketing and product customization for improving the MaaS transport environment. They can also be used to lay a foundation for traffic balance under the MaaS system. First, end users should not be regarded as a homogeneous group. The travel-chain complexity environment has different effects on the public transport travel intentions of different travelers. When building a traveler-oriented MaaS system, it is necessary to classify user types to better formulate service plans for each part of the population [54]. Second, when designing a MaaS system, a special line for children traveling to and from school, or a special line for families traveling with children, should be established with a higher service level to attract more travelers to using public transport. Additionally, in the situation in which men have lower tolerance for the travel-chain complexity environment, more male MaaS passenger sharing lines can be designed to reduce transfers. Finally, for those who do not own cars, green travel should be highly publicized. Thus, they will be affected by not only their preference for public transport but also the subjective norms of green travel, and they will be less likely to transfer to private cars. 

## 8. Directions for Future Research

In the future, additional research will be conducted on public transport travel intention based on travel chains and service quality in combination with MaaS and SaaS (software as a service). This paper is mainly based on the data of the questionnaire. If we can utilize the real-time online GPS data on the MaaS platform to study the travel intention, the conclusions may be richer and more accurate in the future study. 

This study’s findings can provide theoretical support for improving the public transport environment and optimizing the travel structure.

## Figures and Tables

**Figure 1 ijerph-20-04547-f001:**
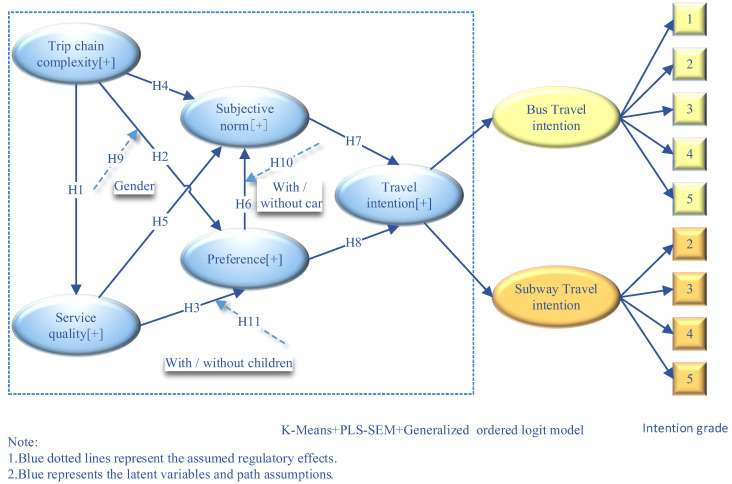
Logic diagram.

**Figure 2 ijerph-20-04547-f002:**
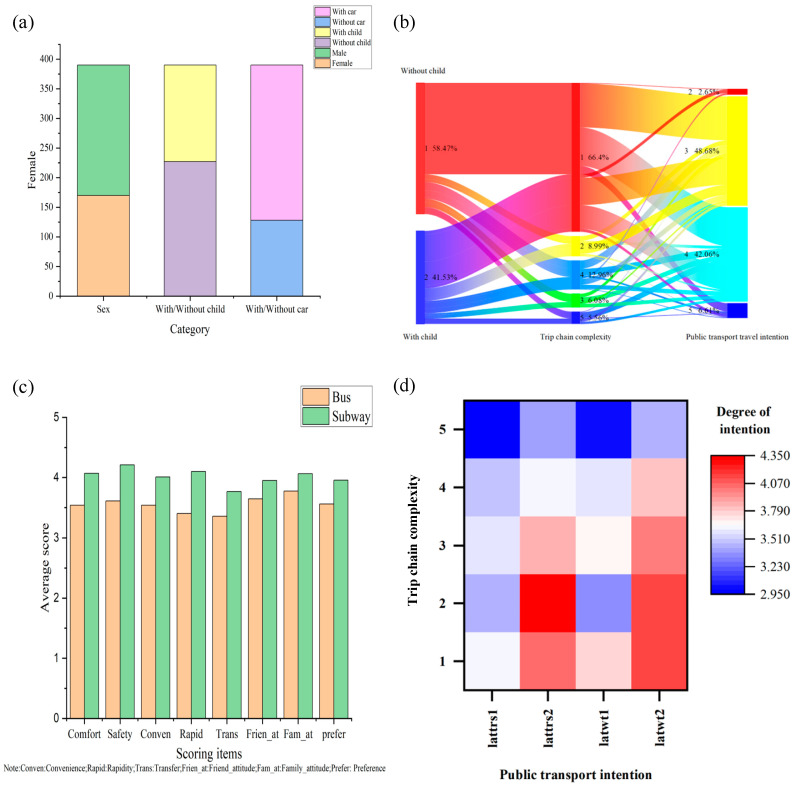
Description of the samples. ((**a**) The heterogeneous samples, (**b**) The effect of travel-chain complexity on public transport travel intention with/without children, (**c**) Respondents’ perceptions of the service quality of different public transport types, (**d**) The change in public transport travel willingness with the increase in travel-chain complexity).

**Figure 3 ijerph-20-04547-f003:**
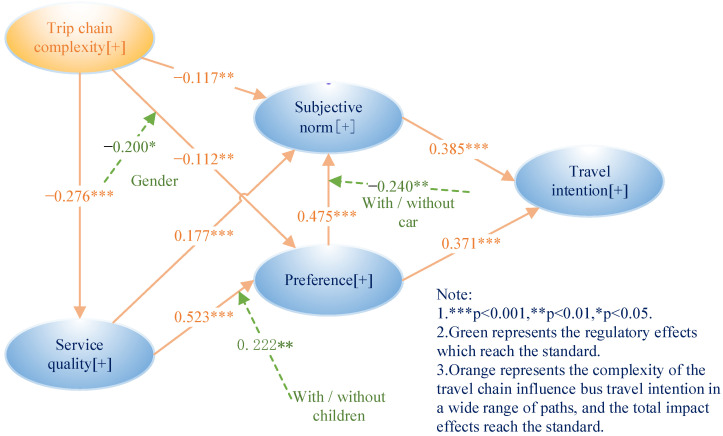
Statistical model diagram.

**Figure 4 ijerph-20-04547-f004:**
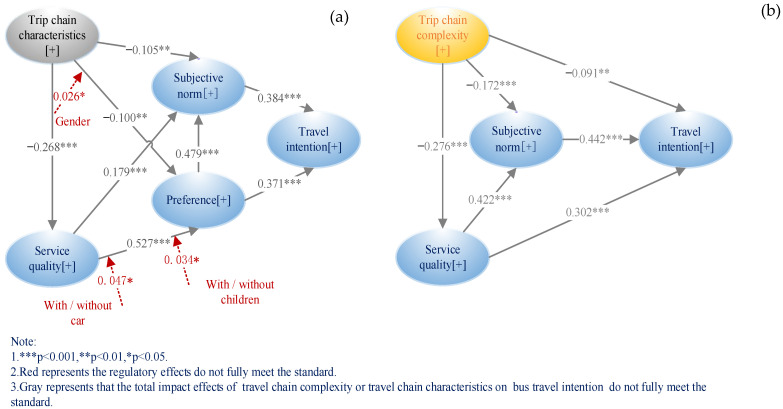
Comparison between Model 1 and Model 2. Note: (**a**) is model1(designed from the perspective of bounded rationality without K-means clustering). (**b**) is Model 2(PLS-SEM only with TPB).

**Figure 5 ijerph-20-04547-f005:**
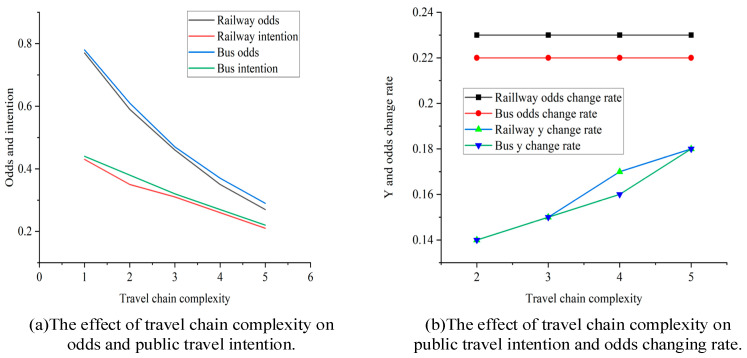
Changes in bus and subway travel intentions as the complexity of the travel chain increases.

**Table 1 ijerph-20-04547-t001:** Operational definitions.

Latent Variable	Definition	Source
Travel-chain complexity	A sequence of trip segments beginning at the “home” activity and continuing until the traveler returns home.	[1]
Service quality	From the perspective of passengers, the overall evaluation of public transport services obtained through objective measurement or subjective perception.	[41]
Preference	Public preference for a certain mode of public transportation.	[26]
Subjective norms	The social pressure an individual perceives urging him/her to implement or not implement a behavior. This pressure mainly comes from external normative beliefs and motivations.	[33]
Intention to use	Travelers’ intention to choose public transportation.	[42,43]

**Table 2 ijerph-20-04547-t002:** K-means clustering results for travel-chain types.

Travel-Chain Type	Travel-Chain Complexity	Number of Cases	*p*
1	1	133	0
2	2	211	
6	3	13	
4	4	20	
7	5	13	
		390	
		0	

**Table 3 ijerph-20-04547-t003:** K-means clustering results for trip times.

Number	Category	Travel-Chain Type	Travel-Chain Complexity	Number of Cases	*p*
1	Very simple	2	1	177	0
2	Simple	5	2	36	
5	General	4	3	139	
4	Complex	1	4	12	
1	Very complex	3	5	26	
3				390	
missing				0	

**Table 4 ijerph-20-04547-t004:** Factor loadings and significance tests.

Latent Variable	Index	Factor Loading (O)	Standard Error (STDERR)	*t*-Test (|O/STDERR|)	*p*-Value
Travel chain	ChSize	0.807	0.044	18.467	***
complexity	activno	0.701	0.052	13.412	***
	tripno	0.799	0.038	21.185	***
Service	Q1cbus	0.695	0.031	20.419	***
quality	Q1csub	0.689	0.030	22.887	***
	Q1sbus	0.722	0.028	25.793	***
	Q1ssub	0.634	0.036	17.478	***
	Q1stbus	0.689	0.038	18.258	***
	Q1stsub	0.591	0.047	12.661	***
	Q1trfbus	0.592	0.044	13.346	***
	Q1vbus	0.600	0.049	12.271	***
	Q1vsub	0.556	0.054	10.304	***
Subjective	eval1	0.794	0.024	33.515	***
norms	eval2	0.809	0.024	34.348	***
	eval6	0.774	0.027	28.771	***
	eval7	0.834	0.022	38.642	***
Preference	habby1	0.833	0.019	43.922	***
	habby2	0.836	0.021	39.251	***
Use intention	lattrs1	0.727	0.033	22.244	***
	lattrs2	0.705	0.032	21.770	***
	latwt1	0.788	0.025	32.105	***
	latwt2	0.771	0.026	29.108	***

*** *p* < 0.001.

**Table 5 ijerph-20-04547-t005:** Reliability, convergence, and discriminant validity analysis.

Latent Variable	α	CR	AVE	1	2	3	4	5
Subjective norms	0.816	0.879	0.645	0.803				
Preference	0.564	0.821	0.697	0.603	0.835			
Use intention	0.737	0.836	0.560	0.609	0.603	0.748		
Travel-chain complexity	0.667	0.813	0.593	−0.288	−0.256	−0.301	0.770	
Service quality	0.824	0.863	0.414	0.472	0.554	0.528	−0.276	0.643

Note: AVE: average variance extracted; CR: composite reliability. Diagonal bold characters are AVE square root; lower triangle is the Pearson’s correlation of latent variables.

**Table 6 ijerph-20-04547-t006:** Internal values of collinearity.

	Subjective Norms	Use	Preference	Travel-Chain	Service
	Norms	Intention		Complexity	Quality
Subjective norms		1.572			
Use intention					
Preference	1.467	1.572			
Travel-chain complexity	1.1011		1.082		1.000
complexity					
Service quality	1.484		1.082		

**Table 7 ijerph-20-04547-t007:** Path significance analysis.

Independent Variable	Dependent Variable	Factor Loading (O)	Standard Error (STDERR)	*t*-Test (|O/STDERR|)	*p*-Value	R2	Q2
Service quality	Travel-chain complexity	−0.276	0.046	6.026	0.000	0.072	0.030
Preference	Travel-chain complexity	−0.112	0.037	3.025	0.003	0.319	0.214
	Service quality	0.523	0.044	11.893	0.000		
Subjective norms	Travel-chain complexity	−0.117	0.036	3.253	0.001	0.404	0.254
	Service quality	0.177	0.047	3.728	0.000		
	Preference	0.475	0.046	10.354	0.000		
Use intention	Subjective norms	0.385	0.051	7.539	0.000	0.458	0.250
	Preference	0.371	0.062	6.009	0.000		

**Table 8 ijerph-20-04547-t008:** Mediation effect analysis.

Path	Factor Loading (O)	Sample Means (M)	Standard Deviation (STDEV)	*t*-Test (|O/STDEV)	*p*-Value
Service quality → Preference → Subjective norms → Travel intention	0.096	0.097	0.017	5.641	0.000
Travel-chain complexity → Service quality → Preference	−0.144	−0.147	0.024	5.976	0.000
Travel-chain complexity → Preference → Subjective norms → Travel intention	−0.020	−0.021	0.007	2.728	0.007
Travel-chain complexity → Service quality → Preference → Travel intention	−0.054	−0.055	0.014	3.913	0.000
Service quality → Preference → Subjective norms	0.249	0.251	0.033	7.503	0.000
Service quality → Preference → Travel intention	0.194	0.197	0.040	4.846	0.000
Travel-chain complexity → Service quality → Subjective norms → Travel intention	−0.019	−0.019	0.007	2.800	0.005
Travel-chain complexity → Service quality → Subjective norms	−0.049	−0.050	0.015	3.300	0.001
Travel-chain complexity → Preference → Travel intention	−0.041	−0.044	0.017	2.463	0.014
Travel-chain complexity → Subjective norms → Travel intention	−0.045	−0.046	0.016	2.822	0.005
Travel-chain complexity → Service quality → Preference → Subjective norms	−0.026	−0.027	0.006	4.217	0.000
→ Travel intention					
Travel-chain complexity → Service quality → Preference → Subjective norms	−0.069	−0.070	0.014	5.032	0.000
Preference → Subjective norms → Travel intention	0.183	0.184	0.030	6.156	0.000
Travel-chain complexity → Preference → Subjective norms	−0.053	−0.055	0.019	2.857	0.004
Service quality → Subjective norms → Travel intention	0.068	0.069	0.022	3.112	0.002

**Table 9 ijerph-20-04547-t009:** Two total-effect tests.

Path	Original Sample (O)	Standard Deviation (STDEV)	*t*-Test	*p*-Value
Service quality → Travel intention	0.358	0.037	9.581	0.000
Travel-chain complexity → Travel intention	−0.205	0.032	9.581	0.000

**Table 10 ijerph-20-04547-t010:** Subway travel intention.

Parameter	B	Exp(B)	Hypothesis Test			Uncentralized Mean	Nominal Variable vs. Nominal Variable
			Wald Chi-Square	df	*p*		
[Subway travel probability = 2]	−6.100	0.004	140.79	1	***		0.649 ***
[Subway travel probability = 3]	−2.462	0.135	66.11	1	***		
[Subway travel probability = 4]	1.039	4.480	14.83	1	***		
Q1ssub_mc	0.526	1.692	7.30	1	**	4.21	
Q1stsub_mc	0.470	1.600	7.01	1	**	4.10	
eval7_mc	1.228	3.416	42.20	1	***	4.06	
habby2_mc	1.073	2.924	32.13	1	***	3.96	
activeno	−0.262	0.770	3.91	1	**	1.76	

Note: ** represent 1% significance level; *** represent 0.1% significance level.

**Table 11 ijerph-20-04547-t011:** Bus travel intention.

Parameter	B	Exp(B)	Hypothesis Test			Uncentralized Mean	Nominal Variable vs. Nominal Variable
			Wald Chi-Square	df	*p*		
[Bus travel probability = 1]	−6.337	0.002	123.55	1	***		0.686 ***
[Bus travel probability = 2]	−4.220	0.015	136.58	1	***		
[Bus travel probability = 3]	−0.811	0.444	10.76	1	**		
[Bus travel probability = 4]	1.906	6.729	52.03	1	***		
Q1vbus_mc	0.307	1.359	3.32	1	*	3.54	
Q1trfbus_mc	0.377	1.458	6.15	1	**	3.36	
eval1_mc	0.296	1.344	3.17	1	*	3.64	
eval6_mc	0.483	1.621	8.41	1	***	3.77	
habby1_mc	1.199	3.317	48.46	1	***	3.56	
ChSize1	−0.250	0.779	4.81	1	**	1.88	

Note: * represents 5% significance level; ** represent 1% significance level; *** represent 0.1% significance level.

**Table 12 ijerph-20-04547-t012:** Path coefficients according to gender.

Path	Female		Male		Difference			
	Estimate	S.E.	Estimate	S.E.	Path-Diff	*t*-Test	Path-Diff	*p*-Values
Travel-chain complexity → Preference	0.001	0.068	−0.199	0.047	−0.200	2.023	−0.200	0.028

**Table 13 ijerph-20-04547-t013:** Path coefficients with/without vehicle ownership.

Path	Without Car		With Car		Difference			
	Estimate	S.E.	Estimate	S.E.	Path-Diff	*t*-Test	Path-Diff	*p*-Value
Preference → Subjective norms	0.616	0.058	0.376	0.065	−0.240	2.386	−0.240	0.006

**Table 14 ijerph-20-04547-t014:** Path coefficients with/without children.

Path	Without Children		With Children		Difference			
	Estimate	S.E.	Estimate	S.E.	Path-Diff	*t*-Test	Path-Diff	*p*-Value
Service quality → Preference	0.419	0.055	0.641	0.053	0.222	2.706	0.222	0.007

**Table 15 ijerph-20-04547-t015:** Comparison of Model 1 and Model 2 (overall evaluation).

Evaluate	Reference Value	Model 1	Ok	Model 2	Ok	Model 3	Ok	VS1	VS2
Index									
1. VIF	0.20 < VIF < 5.00	1.00 < VIF < 1.57	√	1.00 < VIF < 1.33	√	1.00 < VIF < 1.57	√		↑ 6.62%
2. R2	R2 ≤ 0.02 weak; 0.02 < R2 ≤ 0.13 middle;	R2 = 0. 46		R2 = 0.46		R2 = 0.49		↑ 7.64%	↑ 6.52%

			√		√		√		

	0.13 < R2 ≤ 0.26 strong.								
3. Q2	Q2 ≥ 0	0.028 < Q2 ≤ 0. 0.253	√	0.043 ≤ Q2 ≤ 0. 0.249	√	0.030 < Q2 ≤ 0. 0.254	√		↑ 2.00%
4. GOF	GOFsmall = 0.100,	GOF = 0.904	√	GOF = 0.902		GOF = 0.947	√	↑ 4.76%	↑ 4.99%
	GOFmedium = 0.250,				√				
	GOFlarge = 0.360.								
5. Path evaluation	|±0.2| ≤ Pathcoefficient	−0.193	!	−0.166	!	−0.203	√	↑ 5.18%	↑ 22.29%

Note: VS1 is the comparison of Model 3 and Model 1; VS2 is the comparison of Model 3 and Model 2. The path evaluation focuses on travel-chain complexity → travel intention; ↑ refers to improvement; √ means that the requirements are met.

**Table 16 ijerph-20-04547-t016:** Comparison of path and regulatory effects between Model 1 and Model 2.

Comparison of Regulatory Effects	Model 1	Model 2	Promote	Standard Source
Regulating variable: Gender				≥± |0.20|
Travel-chain characteristics	−0.026 *	−0.200 *	↑ 669.23%	
/ Travel-chain complexity→				[51]
preference				

Notes: ↑refers to improvement; * represents 5% significance level.

**Table 17 ijerph-20-04547-t017:** Effect of travel-chain complexity on subway travel willingness when other variables are mean values.

Travel-Chain Complexity (Activeno)	1	2	3	4	5	Pearson’s Correlation
ln(y/(1 − y) = (−0.262) * activeno	−0.262	−0.524	−0.786	−1.048	−1.31	0.683 ***
Travel intention level (PLS-SEM)	4	4	4	4	4	
*odds* = y/(1 − y)	0.7695	0.5921	0.4557	0.3506	0.2698	
*odds* ratio = *oddsi*/*oddsi* + 1		1.2996	1.2993	1.2997	1.2994	
(*oddsi*-*oddsi* + 1)/*oddsi*		0.2305	0.2304	0.2306	0.2305	
y	0.4349	0.3519	0.3130	0.2596	0.2125	
△y		0.0830	0.0389	0.0534	0.0471	
(yi − yi + 1)/yi		0.1449	0.1538	0.1706	0.1814	

Note: * represents 5% significance level; *** represent 0.1% significance level.

**Table 18 ijerph-20-04547-t018:** Effect of travel-chain complexity on bus travel willingness when other variables are average values.

Travel-Chain Complexity (Chsize1)	1	2	3	4	5	Pearson’s Correlation
ln(y/(1 − y) = (−0.262) * activeno	−0.250	−0.500	−0.750	−1.000	−1.250	0.608 ***
Travel intention level (PLS-SEM)	4	4	4	3	3	
*odds* = y/(1 − y)	0.7788	0.6065	0.4724	0.3679	0.2865	
*odds* ratio = *oddsi*/*oddsi* + 1		1.2787	1.2979	1.2703	1.2759	
(*oddsi*-*oddsi* + 1)/*oddsi*		0.2212	0.2211	0.2212	0.2212	
y	0.4378	0.3775	0.3208	0.2690	0.2227	
△y		0.0603	0.0567	0.0518	0.0463	
(yi − yi + 1)/yi		0.1377	0.1502	0.1615	0.1721	

Note: * represents 5% significance level; *** represent 0.1% significance level.

## Data Availability

Some or all data, models, or code generated or used during the study are available from the corresponding author by request.

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
