# Peer review of "The Effect of Travel-Chain Complexity on Public Transport Travel Intention: A Mixed-Selection Model"

_ijerph, 2023, doi:10.3390/ijerph20054547_

Round 1

Reviewer 1 Report

Dear authors,

I am glad I have this opportunity to review your manuscript.  In this paper, The research aim is to explore the effects of travel-chain complexity environment on public transport intention by a mixed selection approach between Partial Least-Squares Structural Equation Modeling and ordered Logit econometric regression.

The title of the paper reflects its content. The research design is appropriate. The methodology is well explained. Possible future research described. The article is written in a transparent way, but the level of scientific rigor is insufficient. My comments/notes are as follows:

INTRODUCTION:

1.       The literature review is neither exhaustive nor up-to-date. There are many recently published studies related to this work that have not been taken into account. Only 2 reference corresponds to studies published during the last 2 years. The authors did not mention a number of important related studies in the literature review that are also much more current. I suggest supplementing the literature review with the following articles:  doi: 10.3390/math9161844;

2.       Line 56. The equilibrium problem of multi-mode public transport arises, for example, in a certain period of time in which buses are overcrowded while subway transport capacity is not saturated. In the 58 long run, such an unbalanced state wasted public transport resources and resulted in low operation efficiency.” This last statement should be justified.

LITERATURE REVIEW:

3.       Line 132. “There is, however, less empirical evidence from the perspective of bounded rationality. Considering the limited literature on that topic, the present study aimed to make a substantive contribution to transportation theory and methodology.”

Line 144. “Few studies, however, have specifically investigated preferences in the travel chain from the perspective of bounded rationality (Liao et al., 2020). This study, therefore, aimed to combine TPB with preference and introduce bounded rationality into travel-chain research, giving consideration to heterogeneity.”

Line 209. “This method introduces SEM results into the Logit model as an effect function. However, it does not check whether all variables in SEM are suitable for the Logit model.) The results of SEM were not compared with those of the Logit model. Therefore, the present study considered these issues.”

The gap of the research has to be defined clearer.Also, different objectives are presented in different parts of the introduction and review of the literature and the final objective of the research is not clearly defined. The objective of the research has to be clearer through specific hypothesis that will be solved by your research in the conclusions section.

MODELING APPROACH:

4.       Line 307. “Referring to Tang and Wu (2016), PLS-SEM analysis was conducted by two stages. First, the measurement model was evaluated to determine whether it had sufficient reliability and validity. Second, the structural model was evaluated to understand the establishment of the research hypothesis.” The objective of the research is not clearly defined. Clear hypothesis should be formulated to be answered directly in the conclusions section.

DATA COLLECTION AND SAMPLE DESCRIPTION:

5.       Line 387. Figure 2(d). "Trip chain complexity" is not correctly oriented.

DATA ANALYSIS AND MODEL EVALUATION:

6.       Line 705. “Figure 4. This is a figure. Schemes follow the same formatting. Comparison between Model 1 and Model 2.”  Mistake in description of Figure 4.

CONCLUSIONS AND DIRECTIONS FOR FUTURE RESEACH:

7.       Line 816. “This study established a mixed selection model to study the travel intention of public transport in the complex travel chain environment and obtained the following conclusions.” The description of the study conclusions should be improved, giving a clear answer to the hypothesis that have to be defined previously. No possible limitations of the study are indicated.

In summary, the research project of the authors is an interesting one. However, additional research works should be performed in my opinion. My recommendation to the authors would be improve the paper in the lines of the above suggestions and to resubmit it.

Author Response

Dear reviewers:

Thank you for your decision and constructive comments on my manuscript. We carefully considered the suggestion of reviewers and made the corresponding changes. Please see the Attachment, in which all comments are carefully responded point-to-point.

Reviewer 2 Report

One more real-time test case study should be incorporated by carrying out the SEM to justify the travel chain for transport.

Author Response

(The authors gave the same response as above.)

Reviewer 3 Report

The article deals with a relevant theme: the need for a multimodal integration for urban mobility.

Here are some points that could be improved:

1 - The title is too long for a single sentence, making it difficult to understand. I suggest its simplification

2 - The transport chain (interconnected multimodal transport) is well explained in the introduction, but between lines 60 and 71, the explanation about MaaS repeats some unreliable articles. MaaS should not be a "goose that lays the golden eggs": to effectively improve urban mobility, MaaS needs to integrate public transport, so that MaaS itself does not solve problems, but it can open up the possibility of desirable intermodality. I suggest re-reading the entire article (including bibliographic review and conclusions) avoiding offering MaaS excessive expectations regarding its positive impacts. Developing countries that only have MaaS for individual transport experience high rates of degradation of mobility encouraged by MaaS services, to the detriment of already deficient public transport.

3 - The TPB is well explained (136 to 147) and is important to reinforce the importance of the research. It can be highlighted in the Abstract.

4- The conclusions can be reinforced, especially by establishing a language for public policy makers in each of the topics with direct indications of findings.

Author Response

(The authors gave the same response as above.)

Round 2

Reviewer 1 Report

I consider that the article only requires minor corrections before its publication. The authors have addressed most of my corrections, except what I detail below:

DATA COLLECTION AND SAMPLE DESCRIPTION:

1. Line 387.Figure 2(d)."Trip chain complexity". The format of this tittle is still wrong. Please review.

 REFERENCES:

 2.    Line 932.  11. Alberto, Romero-Ania, Lourdes, Rivero Gutiérrez, L., María Auxiliadora, De Vicente Oliva. (2021). Multiple Criteria Decision 932 Analysis of Sustainable Urban Public Transport Systems. Mathematics, 9(16). doi:10.3390/math9161844

Reference 11 is wrong. This is the correct way to cite it:

Romero-Ania, A., Rivero-Gutiérrez, L., De Vicente-Oliva, M.A. (2021). Multiple Criteria Decision 932 Analysis of Sustainable Urban Public Transport Systems. Mathematics, 9(16). doi:10.3390/math9161844

Author Response

Dear reviewers:

Thanks for your further comments on my manuscript. We revised carefully the paper to ensure its quality. I hope the paper is clear in the second round. Could you please see the Attach?
